

# Dynamic boreal summer atmospheric circulation response as a negative feedback to Greenland melt during the MIS-11 interglacial

Brian R. Crow[1], Matthias Prange[1], Michael Schulz[1]

[1]MARUM (Center for Marine Environmental Sciences) and Faculty of Geosciences, University of Bremen, Bremen, 28359, Germany

*Correspondence to*: Brian R. Crow (bcrow@marum.de)





**Abstract.** The unique alignment of orbital precession and obliquity during the Marine Isotope Stage 11 (MIS-11) interglacial produced perhaps the longest period of planetary warmth above pre-industrial conditions in the past 800 kyr.

Reconstructions point to a significantly reduced Greenland ice sheet volume during this period as a result, although the remaining extent and volume of the ice sheet are poorly constrained. A series of time-slice simulations across MIS-11 using a coupled climate model indicates that boreal summer was particularly warm around Greenland and high latitudes of the Atlantic sector for a period of at least 20 kyr. This state of reduced atmospheric baroclinicity, coupled with an enhanced and poleward-shifted intertropical convergence zone and North African monsoon, favored weakened high-latitude winds and the

emergence of a single, unified midlatitude jet stream. Consequent reductions in lower-tropospheric eddy heat flux over the north Atlantic therefore emerge as a negative feedback to additional warming over Greenland, perhaps partially counteracting conditions otherwise very favorable for widespread melting of the ice sheet. The relationship between Greenland precipitation and the state of the North Atlantic jet is less apparent, but slight summer changes in precipitation appear to be more than offset by increases during the remainder of the year. Such a dynamic state is surprising, as it bears

stronger resemblance to the unified-jet state postulated as typical for glacial states than to the modern-day interglacial state.

## 1 Introduction

The Marine Isotope Stage 11c interglacial (approximately 424 ka to 395 ka; hereafter MIS-11) is the longest and one of the warmest interglacials of the past million years (e.g., Lisiecki and Raymo, 2005; Raymo and Mitrovica, 2012). Its climatological significance lies in the extent to which sea levels rose during this period, estimated at around 6-13 meters

above that of the present day (Dutton et al., 2015). A substantial percentage of this rise is attributed to the melt of the Greenland ice sheet (GrIS), which may have contributed as much as 4 m to 7 m of its estimated 7.4 m of present-day sea-level equivalent water content (Morlighem et al., 2017; Robinson et al., 2017). Antarctica has recently been estimated to have contributed another 6.7-8.2 m (Mas e Braga et al., 2021). The prolonged warmth of this period therefore is of direct relevance to understanding the processes that cause GrIS melt, a highly pertinent question as planetary warming is likely to

continue in the near future.

As suggested by the wide range in sea-level rise estimates, considerable uncertainty exists with regards to both the degree of melt of the GrIS and the global and regional temperature anomalies during MIS-11. Pollen records indicate the development of some boreal coniferous forest around the margins of southern Greenland at some point during MIS-11 (de Vernal and Hillaire-Marcel, 2008; Willerslev et al., 2007), indicating both the prevalence of ice-free ground and sufficient summer

warmth to support tree growth. Peak temperatures during this time remain poorly constrained, however. While some ice-core data (e.g., Masson-Delmotte et al., 2010) and SST reconstructions (Dickson et al., 2009) suggest global temperatures 1-2°C warmer than pre-industrial, with Arctic anomalies potentially several degrees higher (Melles et al., 2012), orbital parameters and greenhouse-gas (GHG) measurements are broadly similar to those of the Holocene (Berger and Loutre, 2003). As such,





climate models have historically struggled to replicate the temperature anomalies implied by both the limited paleo-
temperature records and the implied degree of GrIS melt (e.g., Robinson et al., 2017; Reyes et al., 2014).

A mechanism invoked as potential means for achieving and sustaining higher temperatures during MIS-11, particularly in
the Arctic during boreal summer, is a strengthened Atlantic meridional overturning circulation (AMOC; Rachmayani et al.,
2017). The authors of that study ascribe this strengthening primarily to salinity increases in the North Atlantic, in turn a
product of favorable alterations of the surface wind and pressure fields inducing stronger ocean surface currents. The extent
to which factors related to atmospheric transports of heat and moisture are involved is left as a question for future research,
and one that we attempt to address further in our analysis.

Multiple modelling approaches have been utilized to attempt to refine understanding of the MIS-11 interglacial climate.
Computing resources are a major limitation to simulating multi-millennial timescales with modern complex climate models,
thereby inducing a choice to use simplified or low-resolution models like Earth system models of intermediate complexity
(EMICs; e.g., Yin and Berger, 2012) or simulating shorter "snapshots" of the climate state at representative key periods (e.g.,
DeConto and Pollard, 2003; Herold et al., 2012; Stone et al., 2013; Milker et al., 2013; Rachmayani et al., 2016, 2017). The
latter approach is often referred to as the time-slice method, and is frequently adopted when utilizing complex coupled
atmosphere-ocean global climate models (AOGCMs). A complex AOGCM such as the Community Earth System Model
(CESM; Hurrell et al., 2013; Gent et al., 2011) would require many months of computation time to complete a transient
simulation of the MIS-11 interglacial even on a high-performance computer; thus the time-slice approach is more practical
for capturing the evolution of the climate over such a long period.

Still others have utilized a combined approach, running both an EMIC and an AOGCM over several time slices to compare
with each other and reconstructions. Kleinen et al. (2014) produced broadly similar climate states with both CLIMBER2 (an
EMIC) and CCSM3 (an AOGCM), though the increased resolution of CCSM3 enabled much better identification of regional
climate features, such as the enhanced African summer monsoon during the 410 ka and 416 ka periods of MIS-11. The
limited spatial and temporal resolution of records and reconstructions during MIS-11 makes verification difficult (e.g. Milker
et al., 2013), but identifying regional climatic changes with teleconnection potential is important to understanding the
climate mechanisms influencing Greenland.

A key aspect of replicating the regional distribution of temperature changes under different climate forcing regimes is
adequately capturing feedback mechanisms internal to the climate system. Orbital forcing in particular has widespread
consequences, as different distributions and intensities of surface heating cause the atmospheric and oceanic circulations to
respond in different ways (e.g., Merz et al., 2015; Fischer and Jungclaus, 2010). Despite relatively modest changes in the
magnitude of seasonal insolation values throughout most of MIS-11, the latitudinal distribution of insolation is still notably
different relative to the pre-industrial period. The high Northern Hemisphere summer insolation during a long interval of
MIS-11 was responsible for both enhancing the African monsoon (Rachmayani et al., 2016; Mohtadi et al., 2016; Wu and
Tsai, 2021) and weakening the mean hemispheric baroclinicity. Both weakened midlatitude baroclinicity in the atmosphere
and enhanced tropical forcing have been identified as mechanisms for shifting the preferred state of the North Atlantic upper





tropospheric jet stream from a split regime (separate subtropical and polar front jets) to a unified hybrid jet regime (Lee and
Kim, 2003; Son and Lee, 2005; Andres and Tarasov, 2019). Altered jet regimes in turn have consequences for the
development and propagation of atmospheric eddies, thus affecting a major source of atmospheric heat at high latitudes (e.g.,
Nakamura and Oort, 1988; Overland and Turet, 1994; Serreze et al., 2007).

Internal climate system mechanics contributing to the pronounced melt of the GrIS during MIS-11 remain largely
unidentified. In the present study, we utilize some of the highest-resolution climate simulations performed to date under
MIS-11 conditions in order to parse these mechanisms further. In particular, what atmospheric changes across the North
Atlantic sector were most consequential for mass balance changes in the Greenland ice sheet? We therefore explore to what
extent insolation-induced changes in the jet stream may have led to feedbacks affecting the poleward transport of
atmospheric heat and moisture.

## 2 Methods

### 2.1 Model configuration

The climate model chosen for this study is the Community Earth System Model (CESM) v1.2.2, a fully-coupled atmosphere-
ocean general circulation model with coupled sea ice, land, and runoff components. The CESM and Community Climate
System Model (CCSM) family has been widely utilized in paleoclimate studies. Our particular configuration utilizes the
Community Atmosphere Model version 5 (CAM5). The land and atmosphere models have an approximate resolution of 1.9°
latitude by 2.5° longitude with 30 sigma-hybrid coordinate vertical layers in the atmosphere. The ocean and sea-ice grids are
comprised of an orthogonal curvilinear grid at nominally 1° resolution with the north pole displaced over Greenland to avoid
singularities in flux calculations in the Arctic Ocean.

Fixed present-day ice sheet topography was assumed for all MIS-11 time-slice experiments, and therefore the land ice model
component was disabled. This is perhaps the largest assumption of these simulations, but enables isolation of the effects of
changing orbital and greenhouse gas (GHG) forcings on the simulated climate. Options for applying different ice sheet
configurations were also severely limited by the sparse Greenland ice-core data extending back through MIS-11, and only a
small number of modelling studies have produced transient reconstructions (e.g., Robinson et al., 2017).

A control run was conducted which adheres to the standards set forth by the Paleoclimate Modelling Intercomparison Project
4 (PMIP4) for a pre-industrial baseline (Otto-Bliesner et al., 2017). This simulation was integrated for 2500 years to enable
full equilibration of the surface climate and quasi-equilibration of deep-ocean temperatures. The experimental MIS-11 runs
were branched from year 1500 of the control integration, with only orbital and GHG parameters altered as described below.

### 2.2 Experimental design and parameters

The "time slice" technique (see e.g., Stone et al., 2013; Rachmayani et al., 2017) was utilized in this study, as transient
simulations of 20 kyr duration or more remain prohibitively expensive in a high-resolution AOGCM. Each selected slice is





integrated for 1000 years at constant forcing conditions, enabling quasi-equilibrium at conditions representative of the
selected time. The 1000 years of each experiment consist of 900 years of spin-up time, then the final century is treated as the
equilibrated period. All results presented are therefore 100-year time series or averages from the end of each simulation. The
time slices were chosen to align approximately with minima and maxima in precession (which has a powerful modulating
effect on the seasonal distribution of insolation) during the warm interglacial period of MIS-11. Intermediate 5 kyr steps
were also selected to ensure fuller coverage of the period of interest. The characteristic parameters of each time slice are
detailed in Table 1.

| Experiment | Precession (°) | Eccentricity | Obliquity (°) | $CO_2$ (ppm) | $CH_4$ (ppb) | $N_2O$ (ppb) |
|---|---|---|---|---|---|---|
| 423 ka | 12.2 | 0.011374 | 23.79 | 268.9 | 652.8 | 284.8 |
| 418 ka | 102.6 | 0.013295 | 24.22 | 273.3 | 677.0 | 272.9 |
| 413 ka | 190.6 | 0.014836 | 24.17 | 273.7 | 705.3 | 273.4 |
| 408 ka | 278.5 | 0.015795 | 23.69 | 280.3 | 726.1 | 279.8 |
| 403 ka | 7.2 | 0.016067 | 23.04 | 279.8 | 675.4 | 285.7 |
| 398 ka | 97.9 | 0.015498 | 22.55 | 276.7 | 623.4 | 285.8 |

**Table 1: Orbital parameters and greenhouse gas values used as fixed inputs into each time slice simulation.**

For each experimental simulation, orbital parameters were calculated following Laskar et al. (2004) at the representative
time period. Greenhouse-gas concentrations were obtained from the European Project for Ice Coring in Antarctica (EPICA)
record, primarily consisting of data from Dome C (Siegenthaler et al., 2005; Luethi et al., 2008). Following the experimental
setup detailed in Otto-Bliesner et al. (2017), a nominal +23 ppb adjustment was applied to Antarctic CH4 values, accounting
for the fact that methane persistently exists in higher concentrations in the Northern Hemisphere during interglacials. GHG
values represent means of a 5 kyr window around the representative time (i.e., 423 ka GHG values are given by a mean of all
values between 425.5 ka and 420.5 ka), which accounts for the inherent uncertainty in GHG values due to short-term
variability (centennial and millennial scale) and analysis techniques.

A sensitivity experiment was also conducted to ensure that the results of the time slice experiments were not dependent upon
initialization. To this end, additional simulations for 418 ka were conducted, branching from year 500 of the pre-industrial
control run and from year 500 of the 423 ka run. A comparison of the equilibrated periods (final 100 years of each
integration) showed no statistically significant differences existed between them.

**2.3 Statistical techniques**

A number of correlation plots are presented in the results section. These present Pearson's r values, 95% confidence intervals
of the r values, and p-values. Pearson's r is calculated with respect to the mean values of given quantities for each particular
time slice. For example, when correlating 2 m air temperature and precipitation for all MIS-11 experiments, the values





correlated consist only of the six mean seasonal precipitation values from the time slices and the six mean seasonal

temperature values from the time slices. The n for these correlations is therefore only six, and the degrees of freedom just

four. In reality, 600 total seasonal values likely have a much greater number of degrees of freedom, although each time

slice's 100 years of data is best classified as a red-noise time series and therefore does not have 100 degrees of freedom

(Thomson, 1982). Significance of correlations presented are therefore conservative.

## 3 Results

### 3.1 Surface temperature response

The surface temperature response is the most obvious effect of the varying orbital and GHG conditions throughout MIS-11

(Fig. 1). Global-mean 2 m air temperatures were at or modestly above pre-industrial levels for each of the 423-408 ka

simulations (dark blue). As expected, the temperature pattern is considerably amplified over Greenland, with positive

anomalies reaching 2-3°C above pre-industrial during the warmest period (413 ka).

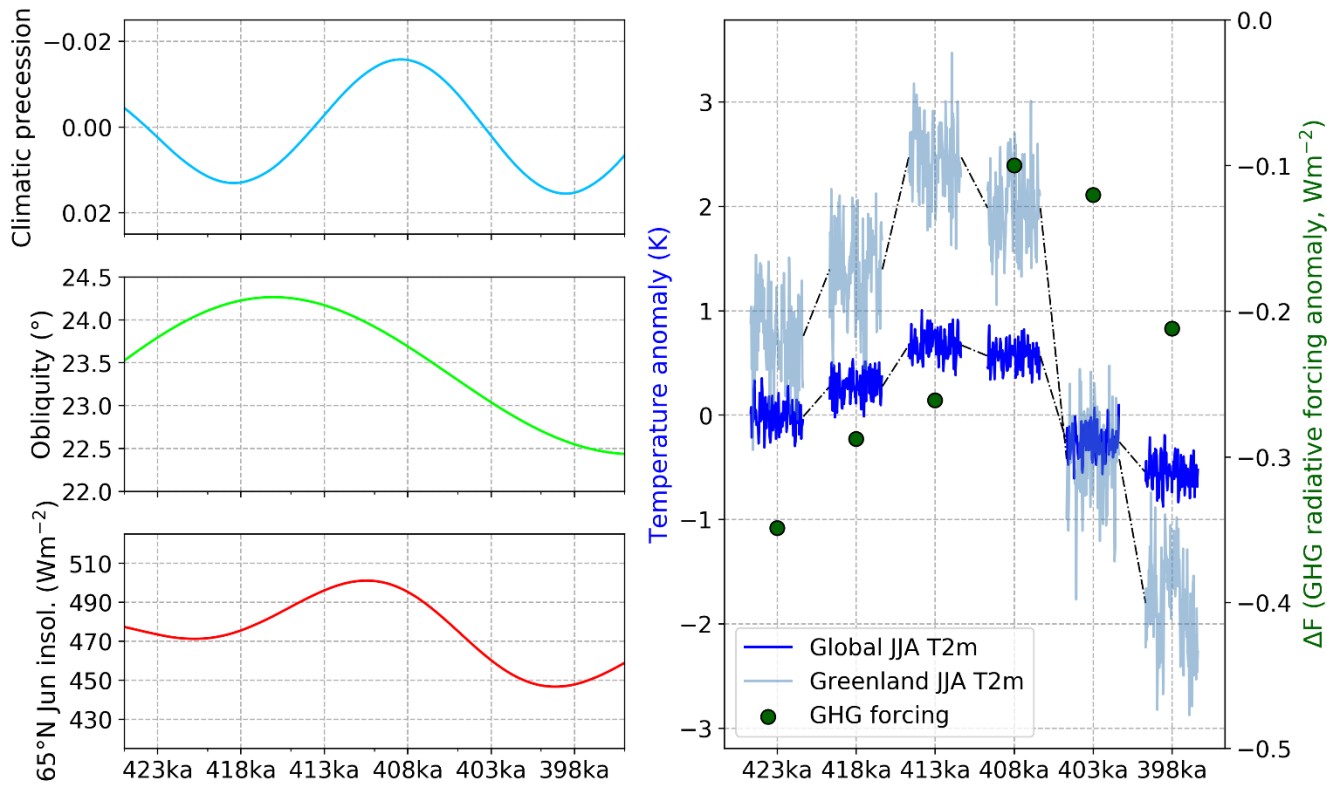


**Figure 1: (Left) Orbital parameters and high-latitude summer insolation during the MIS-11 interglacial. (Right) Mean boreal summer global and Greenland-mean (55-85°N, 280-350°E) 2 m air temperature anomaly "pseudo-time series" relative to the pre-industrial control simulation. The green dots in the right panel indicate the radiative forcing anomaly based on the combined effects of $CO_2$, $CH_4$, and $N_2O$, after IPCC (2001). The pseudo-time series depict seasonal-mean boreal summer (June-July-**





**August; JJA) values from the final 100 years of each simulation; the time x-axis is therefore not to scale and the discontinuities are larger than depicted.**

The spatial distribution and magnitude of boreal summer temperature anomalies across the North Atlantic sector also varied notably throughout the analysis period (Fig. 2). Anomalies as high as 4-5°C above pre-industrial are evident around northern Greenland and the Canadian Arctic during the peak warmth of 413-408 ka due to the combined effects of high insolation and

regional feedbacks, including reduced sea-ice cover. More modest and relatively uniform warming is present across much of the rest of the Atlantic basin, with the exception being areas immediately surrounding the Mediterranean Sea. A narrow but stark band of 1-2°C cool anomalies stands out in sub-Saharan Africa during the 423 ka and 413-408 ka simulations, the result of increased cloud cover and surface cooling induced by evaporation associated with enhanced monsoonal and intertropical convergence zone (ITCZ) convection. Also evident is the abrupt return of surface air temperatures to near-pre-

industrial conditions by 403 ka and substantial cool anomalies by 398 ka, consistent with conditions potentially favorable for renewed glaciation.



**Figure 2: Mean JJA 2 meter temperature anomalies for the final 100 years of each CESM simulation corresponding to the listed timeslices. The green box in the 423 ka panel indicates the averaging region for the mean temperature values used in correlation plots, 55-85°N and 280-350°E. The mean anomaly value listed beneath each panel is a cosine-weighted mean value of the complete shaded area (0-87°N, 270°E-40°E).**




One clear consequence of stronger warming at high latitudes, especially when paired with tropical surface cooling over Africa, is the reduction of the mean equator-to-pole surface temperature gradient, and thus the contribution of the thermal wind balance to the geostrophic flow. A dynamic adjustment of baroclinic processes is therefore to be expected, including 165 attendant changes in jet stream and baroclinic eddy behavior.

## 3.2 Atmospheric circulation response

Particularly robust changes are present in the lower-tropospheric eddy heat flux (EHF) anomalies across the North Atlantic sector. Figure 3 illustrates the mean patterns of total (transient plus stationary) boreal (sensible) summer eddy heat flux anomalies relative to the pre-industrial control simulation. Stationary eddy heat fluxes are defined as $\overline{v^*T^*}$, where $\overline{v^*}$ and $\overline{T^*}$ 170 are the zonally-anomalous meridional wind and temperature, respectively, with anomalies calculated at daily intervals and at the 700 hPa atmospheric pressure level before being averaged to monthly values. Transient eddy heat fluxes are likewise defined as $\overline{v'T'}$, or the product of the mean time-anomalous meridional wind and temperature. The total EHF is simply the sum of the transient and stationary eddy heat fluxes and is, in effect, representative of the heat-advecting effects in the lower-middle troposphere of the atmospheric waves captured in the model. The values represented in the figure are averages of all 175 the June-July-August (JJA) days in the 100 year period. Over the entire analysis domain, total EHF anomalies are dominated by the transient eddy component (not shown).





**Figure 3: Mean lower-tropospheric (700 hPa) total eddy heat flux anomalies over the North Atlantic. Values shown are boreal summer means over the final 100 years of each designated simulation. The green box in the 423 ka panel shows the area over which area-mean values are computed for correlations.**




Two features in the total EHF anomaly fields stand out. First, the couplet of positive and negative anomalies over southwestern Europe and the central Mediterranean during 423-408 ka is indicative of a robust shift in the favored storm track. Positive EHF anomalies are indicative of either an increased and anomalous meridional temperature gradient or frequent large southerly wind anomalies (transient or stationary), or more likely, some combination thereof. Given that the mean temperature gradient appears unchanged or even slightly weaker (Fig. 2), the likely source of this anomaly is increased frequency and/or intensity of wave activity and meridional transport in this region. Conversely, reductions exist over the central Mediterranean, consistent with a shift in baroclinic wave activity away from this region. A second region of widespread reduction in total eddy heat fluxes is present over much of Greenland and the open North Atlantic. The magnitude of this shift is less significant than the anomalies over Europe and the Mediterranean, but still implies a reduction in wave activity and/or intensity, a tendency towards weaker warm advection, or both.

Winter (DJF) — r=-0.858 (-0.977,-0.326), p=0.029

Summer (JJA) — r=-0.97 (-0.996,-0.817), p=0.001

**Figure 4: Correlations between the mean surface air temperature around Greenland and the seasonal mean total (stationary + transient) eddy heat flux over the North Atlantic. Correlations, p-values, and trendlines are calculated using mean values of temperature and EHF anomalies for each of the six timeslices. Individual years are lightly colored small dots, and the mean of each of the six timeslice simulations are given by bold, outlined dots.**



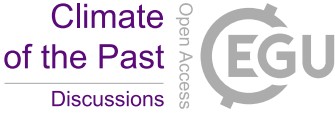

As implied by the eddy flux maps, a strong negative correlation exists between the temperature averaged over Greenland and its immediate surroundings (the region enclosed by the green box in the top-left panel of Fig. 2) and the mean eddy heat flux over the North Atlantic region (the larger region enclosed by the green box in Fig. 3). The strength of this relationship is also

strongly dependent on the season (Fig. 4). While both boreal winter-mean and summer-mean EHF-temperature relationships pass significance at 95% confidence, the correlation is much stronger in boreal summer and with a much-reduced 95% r-value confidence interval. This can be partly explained by the much larger magnitude and variability of seasonal-mean North Atlantic eddy heat fluxes in winter, but is also indicative of a more robust dynamic response in the summer season.

**Figure 5: Changes in the frequency of the latitude and peak velocity of JJA daily maximum 300 hPa winds over the North Atlantic (20-87°N, 90°W-0°E). Latitude bins are approximately 3.8°, containing two model grid cells each. Velocity bins are at intervals of 5**



**m s⁻¹. Frequency matrices are computed relative to the pre-industrial control simulation, and the velocity maxima reflect the maximum single value at any grid point in the analysis domain.**

### 3.3 Jet stream response

While reduced meridional temperature gradients play a role in reducing eddy heat fluxes, changes in eddy activity are the dominant cause. Eddy activity in the midlatitudes has a symbiotic relationship with the jet stream, which both drives and is driven by wave activity. An attendant change is therefore expected in jet behavior, which is apparent in the daily jet-frequency matrices (Fig. 5). A clear shift in the favored latitude and strength of the boreal summer North Atlantic jet is evident, with weaker and lower-latitude daily maximum 300 hPa winds clearly favored in the 418 ka, 413 ka, and 408 ka

experiments. These three warm periods also show a slight reduction in the frequency of low-latitude jet maxima, denoted by the slight negative frequency anomalies on the lower flank of the robust couplet. This is consistent with a decreasing tendency to have wind maxima in the typical (modern) latitude of the subtropical jet, an observation which is further supported by the presence of negative mean 300 hPa wind anomalies (Fig. 6).







**Figure 6: Mean JJA wind anomalies at 300 hPa across the MIS-11 simulations (colors) and climatological 300 hPa wind velocities from the piCtrl simulation (black contours). Evident reductions in mean winds across both the typical subtropical jet location (~25-35°N) and the typical polar jet location (~50-70°N) are present in most time slices, along with increases in mean winds across the eastern midlatitude North Atlantic and western Europe. These features are consistent with a favored unified jet state (Son and Lee 2005).**



Thus, both purely eddy-driven high latitude jet maxima and purely subtropical jet maxima are reduced (the southern flank subtropical wind weakening is even more clearly visible in the u-component of the wind field; not shown), with the dominant state instead emerging as a broad hybrid eddy-thermal jet in the middle latitudes. This result is a confirmation of the observation made by Son and Lee (2005), who demonstrated that either weakened midlatitude baroclinicity or stronger tropical ascent could cause the jet to transition to a unified state. Both mechanisms are present during MIS-11 boreal

summers. Baroclinicity changes are clear in the temperature anomaly patterns (Fig. 2) and implied by the robust negative correlation between jet strength and mean 2 meter air temperature over the North Atlantic (Fig. 7). Meanwhile, negative temperature anomalies and greatly increased precipitation across the ITCZ corridor of the North Atlantic and sub-Saharan African (see Fig. 11) are indicators of significantly enhanced convective activity (and thus tropical ascent) in this sector.



**Figure 7: The relationships between the JJA-mean North Atlantic 2 m air temperature and the latitude (left) and magnitude (right) of maximum 300 hPa winds across the North Atlantic. As in Figure 4, correlations, best-fit lines, and p-values are calculated with respect to the overall means of each timeslice (indicated by bold, outlined dots). The smaller dots represent the seasonal-mean values for each of the 100 years of each time slice (n=600 in total).**



### 3.4 Eddy-jet relationship

Notably, the region with the large couplet in EHF anomalies over western Europe and the Mediterranean (Fig. 3) corresponds with the largest changes in jet stream wind strength (Fig. 6). Since the changes in EHF are dominated by the transient component, it is clear that the emergence of these anomalies is associated with a significantly altered North Atlantic storm track. A much more baroclinically active JJA storm track across this region appears to be the consequence of the merged jet state, which is also consistent with the trapping of atmospheric waves equatorward of a merged jet (e.g.,

Nakamura and Sampe, 2002). Reduced baroclinic eddy growth on the poleward flank of the merged jet explains the existence of the opposite state (reduced EHF) across much of the open North Atlantic. The contrast is made clear from the comparison of jet strength and total eddy heat flux (Fig. 8). At 413 ka, large reductions in EHF are present across the central and northern Atlantic in association with the weakened polar jet; in cooler-than-pre-industrial 398 ka, these have been replaced by small positive anomalies in association with the return to a split-jet state.





**Figure 8: Comparison of the mean boreal summer upper-tropospheric anomalous wind speed (colors) and lower-tropospheric total eddy heat flux (black contours) between the globally warmest MIS-11 simulation (413 ka) and the coldest (398 ka).**



Interestingly, correlations between EHF anomalies and jet characteristics fail to paint quite so clear of a picture. While hints
of a relationship between jet latitude and North Atlantic-mean EHF (mean over a box spanning 40-80°N and 70°W-0°E)
exist, the correlation does not reach significance at the 95% confidence level (Fig. 9). However, EHF exhibits a strong
positive correlation with boreal summer jet strength in the North Atlantic. Given the association our simulations demonstrate
between larger jet wind maxima and the split-jet state, this would imply greater EHF during split-jet regimes, and reduced
EHF during the prevailing merged-jet regime of MIS-11 boreal summers.


**Figure 9: Correlations between summer mean seasonal values of the latitude of maximum 300 hPa winds (left panel), the
magnitude of jet maximum winds (right panel), and total eddy heat flux over the North Atlantic. Correlation does not reach 95%
significance for jet location, but the correlation with jet strength is robust.**

### 3.5 Precipitation and the storm track

With such a pronounced change in eddy behavior as a result of the shifted jet, a logical consequence would seem to be a
positive correlation between jet strength and precipitation over Greenland. However, exactly the opposite appears to be the
case (Fig. 10), with a significant but small negative trend in precipitation associated with higher North Atlantic wind





maxima. As is the case with the EHF correlations (Fig. 9), only the relationship between maximum jet velocity and precipitation obtains 95% significance.



**Figure 10: Relationships between JJA-mean latitude of maximum 300 hPa winds (left), the magnitude of the maximum jet wind (right), and seasonal mean precipitation over Greenland (land-only). As with eddy heat fluxes, the correlation is only significant for jet strength.**

Competing effects appear to be influencing precipitation over Greenland in these simulations: intuitively, increases in eddy heat flux should correspond to increases in eddy moisture flux (EMF). However, comparing the jet state to total EMF over the North Atlantic directly results only in a statistically insignificant correlation (not shown). On the other hand, we have already demonstrated that warmer North Atlantic temperatures are associated with weaker jets, and lower-tropospheric air temperature places a strong cap on atmospheric moisture. The apparent negative correlation between maximum jet winds and precipitation over Greenland (Fig. 10) therefore appears to be overwhelmingly driven by the inverse relationship between temperatures over Greenland and jet strength. Behavioral changes in the jet and eddies exhibit only a secondary influence.



Spatially, the largest boreal summer precipitation changes across the North Atlantic sector appear related to the strength of the African monsoon (Fig. 11). Specifically, a drying is noted across much of the midlatitude Atlantic and much of central and southern Europe, consistent with the general shift of the preferred storm track and suggestive of increased subsidence from the poleward flank of Hadley cell. The magnitude of subtropical drying appears to be roughly proportional to the increase in precipitation over sub-Saharan Africa, strongly suggesting an influence for the tropical convection upon the altered jet state. Precipitation rate increases across west/central Africa associated with the increased monsoon are evident through the 423-403 ka periods, peaking at up to 1.5 to 2 mm day$^{-1}$ under 408 ka conditions. Attendant decreases of up to 0.5 mm day$^{-1}$ over Europe and 1.0 to 1.5 mm day$^{-1}$ across the Caribbean and western subtropical Atlantic are also evident during 413 and 408 ka.

Over Greenland, boreal summer precipitation changes during the warm 423-403 ka period resemble that of orographic precipitation during dominant westerly/northwesterly wind regimes: positive precipitation anomalies along the western slopes and negative anomalies along and just off the southeastern coast. The anomalies are relatively small, on the order of 0.1-0.3 mm day$^{-1}$. However, the mean modelled precipitation over Greenland is just under 2 mm day$^{-1}$, so these changes are potentially significant (though values in coastal regions and along terrain gradients are considerably higher). Coupled with the negative eddy heat flux anomalies, this precipitation pattern would appear to support the notion that baroclinic wave activity during boreal summer is reduced in the vicinity of Greenland, with background westerlies becoming the prevailing pattern.

In the annual mean, however, a near-opposite pattern emerges in precipitation anomalies, particularly for southeastern Greenland (Fig. 12). Simulated annual average precipitation rates increase there by 0.1-1.0 mm day$^{-1}$ for the 418-408 ka period. Since this precipitation increase occurs in the cool season, it is overwhelmingly snow (not shown but confirmed by model precipitation-type output), and therefore contributes to an increase in mass balance of the ice sheet across southeastern regions.





**Figure 11: Timeslice mean summer precipitation anomalies across the North Atlantic throughout MIS-11, indicating slight increases in precipitation on the western slope of Greenland with corresponding decreases in the southeast from 423-408 ka.**


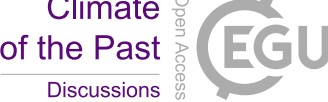

**Figure 12: As in Figure 11, but for the annual mean period.**



## 4 Discussion

Our results compare favorably with the limited previous modelling studies conducted to assess MIS-11 climate. Kleinen et al. (2014) examined key periods of MIS-11 climate using both an EMIC (CLIMBER2) and an AOGCM (CCSM3), finding both the pronounced high-latitude warming and narrow region of tropical cooling that we have identified here with CESM1.2 in both of their models. They similarly identified the enhancement of the African monsoon and the migration and strengthening of tropical convection over the eastern Atlantic, albeit with a more limited spatial extent of increased

precipitation than in our simulations. Further analysis of the same two CCSM3 time slices in Rachmayani et al. (2016) confirmed these temperature and precipitation relationships.

Consensus among recent climate modelling studies therefore indicates a robustly strengthened African monsoon during MIS-11 in boreal summer. Despite severe limitations in the availability and spatial coverage of reliable proxy records from this time period, the data that exists appears to support this. Decreased deposition of terrigenous iron in the deep ocean off the

coast of northwestern Africa during the period 420-396 ka indicates a substantially less dusty (i.e., wetter) northwest African climate during this time (Helmke et al., 2008). Also coincident with this development was the occurrence of increased deposition of organic matter in the eastern Mediterranean (sapropel 11), approximately dated to 407 ka and likely the result of increased freshwater input from the Nile River as eastern African rainfall increased (Kroon et al., 1998; Lourens, 2004). Timing the African wet period to a resolution greater than a few thousand years or determining precisely the magnitude of

the rainfall changes is still beyond the precision afforded by these records; however, they do offer some general verification of the signal produced by the model consensus.

Primary responsibility for the changes in both the African monsoon and the midlatitude baroclinicity naturally lies with the insolation changes. Both inter-hemispheric and intra-hemispheric changes to the insolation gradient have roles to play, with the relatively high obliquity conditions of the early-mid MIS-11 interglacial (ca. 423-408 ka) a chief contributor to the low-

baroclinicity conditions in boreal summer. High obliquity conditions are responsible for increased high-latitude insolation in the summer hemisphere, thus driving the reduction in meridional temperature gradient seen in our simulations (Mantsis et al., 2014). Furthermore, high obliquity enhances the inter-hemispheric temperature gradient, sustaining a stronger Hadley circulation and causing a poleward shift in the Hadley cell in the summer hemisphere (Mantsis et al., 2014). Climatic precession is known to have the dominant effect on the state of the north African monsoon (e.g., Bosmans et al., 2015),

which is illustrated in our simulations by the emergence of the strongest monsoon conditions during 408 ka (and, conversely, the weakest monsoon precipitation anomalies nearest precession maxima at 418 and 398 ka). The favorable alignment of moderate-low precession and moderate-high obliquity throughout much of MIS-11 thus produces sustained Northern Hemisphere boreal summer warmth as well as monsoon and jet responses over an anomalously long period.

As others have already noted, the changes in baroclinicity seen during MIS-11 will likely be mirrored to some extent by

present and future GHG-induced global warming, suggesting the possibility of a strengthened African monsoon and northward-shifted Atlantic ITCZ in the not-so-distant future (Mohtadi et al., 2016, and references therein). In the present



climate, it has been demonstrated that an improved understanding of western African convection can significantly improve weather forecast skill across Europe and the North Atlantic due to teleconnection patterns (e.g., Pante and Knippernitz, 2019). Long-term and large-scale mean circulation anomalies in the west African monsoon region thus have far-reaching

implications for regional or global weather and climate patterns, including over Greenland. Improved understanding of the two-way dynamic linkages between patterns of tropical convection during boreal summer and changes in the North Atlantic circulation are necessary in order to optimize predictions of future Greenland ice sheet behavior.

It also remains to be determined to what extent the jet-eddy response identified here is robust to all models, given the variable degree to which the North Atlantic storm track tends to be southward-biased in Climate Model Intercomparison

Project (CMIP) models (Harvey et al., 2020). Numerical models can also struggle to replicate the magnitude and latitudinal extent of the west African monsoon circulation, which, given the sensitivity of the midlatitude jet to tropical forcing in our simulations, suggests another potential bias of concern (e.g., Brierley et al., 2020). Our study also does not test the sensitivity of the jet-eddy response to varying levels of GHGs under identical orbital forcings, which are subject to some uncertainty for paleo-periods like MIS-11 given millennial-scale fluctuations and the relatively low temporal resolution of GHG records

(variable between a few hundred and approximately 1000 years). Natural limitations on the availability of quality reconstructions will almost certainly continue to prevent true "validation" of modelling results as well.

The use of a fixed modern-day Greenland ice sheet in our climate simulations represents probably our most significant assumption, although the effects of this are minimized during the summer season. Elevation reductions on the order of several hundred meters across the GrIS would result in 2 m air temperature increases of several degrees Celsius, further

enhanced by albedo reductions due to the melting of the ice sheet and emergence of bare rock and soil across marginal regions. In fact, modelling sensitivity studies examining future scenarios have identified these factors as contributing to notably larger GrIS ablation areas and sea-level rise from Greenland melt when accounted for in a two-way coupled simulation (Le clec'h et al., 2019). In contrast, it is possible that these feedbacks may result in further reduced baroclinicity across the North Atlantic sector, possibly further reinforcing the unified-jet state and leading to additional reductions in eddy

heat fluxes. Competing effects on precipitation are also possible: increased atmospheric moisture content is expected over a warmer, lower-elevation Greenland surface; however, the diminished topography in large sections of Greenland would be expected to reduce the production of orographically-induced precipitation. We plan to investigate the dynamic atmospheric response to a greatly-reduced GrIS under MIS-11 climate conditions in a future sensitivity study, and thus hope to better assess the appropriateness of the fixed ice sheet assumption.

## 370  5 Conclusions

Using CESM time-slice simulations of only insolation and GHG forcing conditions corresponding to the MIS-11 interglacial, we have uncovered a robust dynamic atmospheric response that includes negative feedbacks to further melting of the Greenland ice sheet. Increased high-latitude insolation due to a favorable alignment of high obliquity and relatively

low-amplitude climatic precession variations across a time interval of around 20 kyr drives pronounced, sustained boreal
summer surface warming across Greenland and the surrounding high-latitude oceanic regions. It remains a future task to
determine if this insolation-driven surface warming alone is sufficient to produce GrIS melt of the magnitude proposed by
previous studies (e.g., Alley et al., 2010; Reyes et al., 2014; Robinson et al., 2017). Resolving this question is complicated by
the two negative feedbacks we have identified in the atmospheric response to MIS-11 interglacial forcing: (i) a transition in
preferred jet stream behavior leading to reductions in poleward eddy heat flux over the Greenland sector, and (ii) increased
annual-mean precipitation over the ice sheet. The mass balance implications of these feedbacks will be explored in a planned
future ice sheet modelling study.

The preferred boreal summer jet response to MIS-11 forcing identified here, driven by the combined effects of weakened
midlatitude baroclinicity and enhanced tropical ascent over the Atlantic sector, is more consistent with the merged-jet state
considered to be characteristic of glacials rather than interglacials (e.g., Andres and Tarasov, 2019; Merz et al., 2015).
Further study, and comparison with other interglacial climate simulations, may clarify whether this is a unique feature of
MIS-11 or typical of strong interglacials. Earlier studies utilizing medium-resolution CCSM3 simulations (Herold et al.,
2012; Rachmayani et al., 2016) suggest that an enhanced African monsoon and warmer high-latitude temperatures are
typical features of late-Quaternary interglacial states, indicating that at least the primary mechanisms behind the jet changes
tend to be present. However, spatial patterns of warming and the degree and duration of monsoon enhancement appear to
vary considerably between interglacials, and it therefore remains to be seen whether the dynamic conditions present in other
interglacials favor the same northern summer jet configuration.

**Author contributions.** This research was performed as part of the PhD studies of lead author Brian Crow, who advised by
authors Matthias Prange and Michael Schulz. Dr. Prange and Prof. Dr. Schulz provided computing resources, guidance for
experimental design, and extensively discussed results. Both also provided considerable constructive feedback and edits to
the manuscript.

**Competing interests.** The authors declare no competing interests that impacted the research process or the development of
this manuscript.

**Acknowledgements.** The authors would like to thank the DFG and the ArcTrain program for furnishing funding and
computing resources for the research undertaken. We are grateful to the Northern German Supercomputing Alliance (HLRN)
for providing access to their extensive computing resources, enabling the lengthy simulations performed for this research to
be done in a timely fashion. Additionally, the lead author would like to thank the Geosystem Modelling group of the
University of Bremen for helpful comments and feedback during various presentations of preliminary results. Additional
helpful feedback and discussions were provided by Lev Tarasov and Heather Andres of Memorial University, Canada, and
their comments are greatly appreciated.



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
