# Peer review of "Dynamic boreal summer atmospheric circulation response as a negative feedback to Greenland melt during the MIS-11 interglacial"

_Climate of the Past, 2021_

## Author Response (AR1)

*Anonymous Review #1*

*Summary*

*The authors are encouraged to apply to all of their plots significance testing and to better discuss the results as specified in the comments below. So I recommend minor to major revision prior to a possible publication in Climate of the Past.*

**Thank you for your detailed comments here. Many of the comments are directed at clarification and extension of discussion points, which will be addressed point-by-point below. The point regarding significance testing is a fair one and will be addressed extensively (see detailed responses to related comments below). The manuscript will also be reviewed for general readability, including simplification or splitting up of lengthy sentences. We hope the changes as addressed below will prove satisfactory to your concerns and result in a better, more readable manuscript.**

*Major and minor comments*

*L15: Please say in which region the jet stream is unified, I guess in the North Atlantic*

**Updated to read "...emergence of a single, unified midlatitude jet stream across the North Atlantic sector."**

*L15-17: The sentence does not read well and is a speculation which should not appear in the abstract, just focus on the main findings here.*

**Rewritten as, "Consequent reductions in lower-tropospheric eddy heat flux over the North Atlantic therefore emerge as a negative feedback to additional warming over Greenland."**

*L32-35: This is a rather long sentence and the authors use two time indicate which is a bit awkward.*

*In general: The is a tendency of the authors to write rather long sentences so I suggest to critically reiterate of this issue to increase the readability of the manuscript.*

**Attention will be given to the length of sentences in the manuscript generally. However, we are unsure what is meant by "two time indicate [sic]" being used in this particular sentence. The following restructuring is proposed: "Pollen records indicate the development of some boreal coniferous forest around the southern margins of Greenland at some point during MIS-11 (de Vernal and Hillaire-Marcel, 2008; Willerslev et al., 2007). This implies the existence of ice-free ground and a period of sufficiently long and warm growing seasons to support tree growth."**

*L43: ascribe -> ascribed*

**Accepted, but the sentence is grammatically correct in either form.**

*L62-63: This part of the sentence is not clear, what is meant by teleconnection potential.*

**This was indeed vaguely phrased, especially for readers without meteorology backgrounds. The following rephrasing is proposed: "Verifying such climatic signals is difficult due to the limited spatial and temporal resolution of proxy records during MIS-11 (e.g., Milker et al., 2013), but are important to identify to the extent possible. Robust regional climatic changes, especially in the tropics, are known to contribute to remote changes in middle- and high-latitude climate via teleconnection mechanisms (e.g., Yuan et al., 2018)."**

*L70: You need to add a reference for the statement on the weakening of the mean hemispheric baroclinicity. Additionally, you need to mention where in the atmosphere the baroclinicity is reduced I guess in the lower part of the troposphere.*

**We will clarify that the reduced baroclinicity is in the lower troposphere. The sentence will also be restructured to offer all references after our statement, as the reduction in mean baroclinicity due to orbital changes is sufficiently self-evident not to require a standalone reference and is touched on by all references given in this sentence.**

*Table 1: I suggest to include also the preindustrial values here.*

**Preindustrial run values will be added to the table.**

*L126-127 and elsewhere in the manuscript: Please use italic for variables like r and p.*

**Accepted.**

*L130: please use italic for n.*

**Accepted.**

*Section 2.3: This is a good description of how the authors applied the correlation analysis. Still I am interested how the significance and the correlation coefficients change if the authors would use the 600 seasonal values to estimate the correlation.*

**Statement added to explain the reasoning for calculating correlation based on the mean values: "On interannual timescales, regional temperatures and eddy heat fluxes are noisy and influenced by a number of different factors. The most prudent comparison is therefore between the long-term mean patterns in each time slice simulation."**

*Fig 1. You need to say That you applied an area weighted average of the "Greenland region". At least this is my guess.*

We did indeed use a latitude-weighted mean over the region illustrated by the green box in Figure 2. This will be clarified in both figure captions.

*General to the results section:*

*As far as I understood it the authors do not change the Greenland Ice sheet, e.g., lower it to a certain extent. I think the authors need to discuss in more details how this specific setting affects the results. Merz et al. (2014 and 2016) showed a strong influence via the lapse rate effect but also of the sea ice distribution to Greenland temperatures during another interglacial period.*

**This limitation of our study is partially covered in the Discussion section. However, we will incorporate the two recommended references and make mention of this limitation earlier in the paper as well (in the Results).**

*Moreover, the authors ignore in their discussion of the results that they use equilibrium simulations and the climate during MIS-11 is transient. I know that equilibrium simulations are the only possibility in such a study, but potential problems need to be discussed, e.g., the equilibrium simulations might be too cold compared to a potential transient simulation, also the biases of the preindustrial simulations shall be mentioned somewhere to give the reader an idea how well model performs, e.g., maybe the model underestimates the AMOC and thus has a cold bias in the North Atlantic which certainly affects the results around Greenland.*

**Equilibrium simulations may indeed have a slightly different outcome than a fully transient one, but the effect depends on the situation. In our case, the 423 ka simulation is warmer over Greenland than the REMBO/CLIMBER2 simulations of Robinson et al. (2017)/Ganopolski and Calov (2011) due to the presence of much greater Greenland and North American ice cover at that time in their simulations. In theory, the later time slices could be subject to a "memory" effect (or lack thereof) in which they adjust more quickly to cooler GHG and orbital conditions than a transient run would (with its warmer oceans and reduced ice cover). However, we believe our methodology here provides ample time for equilibration of the surface climate, thus minimizing these effects.**

**Each time slice is "spun-up" for 900 years prior to the 100 year "equilibrium" period. That this length of spinup time is sufficient is evidenced by a complete lack of air temperature, sea ice, and upper ocean temperature trends in model time series for the last few centuries. Of course the spin-up time is inadequate for full equilibration of the deep oceans (e.g., Varma et al., 2016), but this is not terribly relevant for surface climate or for effects on the Greenland ice sheet, and drift generally appears to be quite small (on the order of 0.05°C per century or less for depths over 1000m). If the study's focus were on Antarctica, arguably much longer equilibration times would be necessary in order to ensure that ocean temperatures at depths relevant to the ice shelves came into equilibrium. We will add discussion of these effects to the methodology and/or discussion sections, but we consider them as minor.**

**Brief discussion of CESM's known biases seems a prudent addition, and we will therefore incorporate that into our results and/or discussion sections. CESM is known to be too cold in present day and preindustrial conditions at high northern latitudes, which can likely be partially attributed to a few dominant circulation features being stronger than observed and partially due to radiative effects (e.g., Wang et al., 2018).**

*Fig. 2 and other figures: Please say to which you compare the MIS-11 simulations. I also suggest to apply a significance test, e.g. a student t-test to all figures of similar style and only discuss the significant anomalies, e.g., my guess is that most of the differences in the 403 ka simulation is not significant in Fig 2.*

**As requested, t-tests were applied to all temperature, eddy heat flux, wind, and precipitation maps. The assumption of normally distributed values for the 100-year time series at each grid point is sub-optimal, particularly for wind and precipitation, but we have enough degrees of freedom in this dataset that t-tests can provide an acceptable metric for significance. However, note that each grid point is necessarily treated as independent for t-testing purposes, and due to the spatial relationships present, the significance is likely somewhat overstated. Nevertheless, t-testing has confirmed that the large majority of results presented pass significance. We will add a detailed description of our t-test methodology in the methods section (section 2). All figures will be updated to the new versions, which include stippling to mark regions of statistical significance.**

*L169-173: The description of the stationary eddy heat fluxes and the transient ones is wrong. Three is a difference between overbar(v\*T\*) and overbar(v\*) times overbar(T\*). The overbar is the time mean and prime means the deviation of the time mean, so that e.g. overbar(v') is zero. Please clarify this. By the way you assess the "meridional" heat flux, so please be more specific about this in the entire manuscript.*

**Stationary fluxes are calculated as overbar(v\*) times overbar(T\*), or the product of the time-mean zonal anomalies. Transient fluxes are calculated as overbar(v'T'), or the seasonal-mean product of the v and T time anomalies calculated at each individual (daily) time step. This will be corrected in the manuscript. References to "eddy heat flux" will be further clarified as "meridional eddy heat flux" in the manuscript.**

*Fig.3: Please chow the mean state of preindustrial as contours in panels. It makes it easier to seem whether the anomaly is a shift or a strengthening or weakening. Also the color scale has too many steps. Do not forget to apply a significance test and double check your results and concentrate only on significant changes.*

**Preindustrial state contours will be added as overlay, the number of colors reduced, and significance testing added as stippled regions.**

*L181-190: I suggest to rather think in how the meridional heat transport is shifted, in the Mediterranean it looks like a reduction. So I suggest to revise this part and be more specific on the anomalies and how they affect the mean state.*

**There are clear instances of eddy heat flux anomalies in Fig. 3 indicating wholesale pattern changes rather than spatial shifts in existing features, such as the negative anomalies across Greenland and the Labrador Sea. Focusing solely on the shifts in EHF is therefore not sufficient. The text already addresses the shift in the robust positive/negative EHF couplet across Europe and the Mediterranean, and specifically addresses the reduction in the Mediterranean. However, this portion of text will be reworded as suggested to make these points clearer.**

*Fig. 4, right panel: Why do you find negative values for some summers of total EHF? On average, cyclones and anticyclones transport heat to the north in the mid latitudes so I would expect only positive values, maybe it is a matter of the region selected.*

**All shaded values in this figure represent the difference in EHF between the given time slice run and the preindustrial climatology. They are thus anomalies. On the whole, net atmospheric heat transport is poleward, and this would be reflected in any average taken around a given latitude band. However, due to prevailing standing wave patterns, it is possible to have small regions of negative EHF. A similar explanation will be incorporated into the text.**

*L202-203: please remove "r-value" as this is puzzling.*

**Rephrased as "...the correlation is much stronger and has a much narrower confidence interval in boreal summer."**

*Section 3.3. There are studies that the southern tip of Greenland steers the most northern position of the eddy driven jet. I just wonder how this your result e.g. in Fig 5 would affect as Greenland remains in the current setting unchanged.*

**This is indeed an interesting point, but we would be limited to speculation as to the effects of this based on our current experiments. It is hoped to perform a sensitivity experiment in the future, with one or more of the time slice simulations repeated with reduced Greenland topography consistent with modelled melt of the ice sheet. This would enable us to draw more effective conclusions as to the various effects and feedbacks of a reduced ice sheet on North Atlantic climate.**

*L228: I am not sure whether this is the best comparison to be made as the publication of Son and Lee assess the behavior in a very idealized model set up, namely an aquaplanet configuration. The configuration of the authors' model is much more complex.*

While there are undeniably large differences between the model configurations of Son and Lee (2005) and our study, the mechanism they identified remains relevant. The aquaplanet study is illustrative of the basic mechanics involved in defining jet locations and regimes. Our purpose here was merely to show that even in a demonstrably more complex system, the behavior they predict still appears to manifest. The sentence will be rephrased for clarity.

*L231-234: This sentence is not well connected to the ones mentioned beforehand. Also the authors refer to Fig 11 before Fig 8, 9 and 10 so either rearrange the figures or move the sentence to the place where Fig 11 is discussed.*

The sentence will be rephrased to only indirectly reference the increased precipitation over sub-Saharan Africa: "As will be discussed further in Section 3.5, the band of tropical cooling over Africa is a result of increased convection, cloud cover, and evaporative cooling, thus indicating enhanced tropical ascent."

*L245: The authors discuss eddy growth, but they can also calculate the Eady Growth Rate (EGR) which is the standard measure and show that this is reduced at the poleward flank. Note that EGR is a combination of hori. temperature gradient and static stability.*

It is a fair point that discussion of eddy growth should probably incorporate EGR analysis. Our intention is not to discuss in detail dynamic changes in eddy growth regimes; rather, we wish to illustrate the shifting spatial patterns of eddy heat flux associated with orbitally-driven baroclinicity changes. We will rephrase our discussion to focus on the shifting location and magnitude of eddy heat flux, as also suggested in the comment about lines 181-190.

*Fig.8 is not necessary as it is a repetition of Fig. 6 and 3*

Figure will be removed.

*L257-259: Please reformulate this sentences, it is a bit unclear.*

Sentence to be rephrased as: "Our simulations indicate that split-jet states are associated with higher absolute maximum jet strengths. Therefore, the strong positive correlation between higher maximum jet velocities and North Atlantic meridional EHF implies greater EHF during split-jet regimes, and reduced EHF during the prevailing merged-jet regime of MIS-11 boreal summers."

*Section 3.5: I think the authors need to include a discussion on the most relevant processes leading to precipitation in Summer over Greenland. So is it extratropical cyclones, convection, orographic lifting?*

**Convection contributes minimally to Greenland precipitation. Precipitation is therefore largely driven by extratropical cyclones, with substantial enhancement of precipitation by orographic features. We will identify an appropriate reference for this and include a statement in the text.**

*Fig.11 and 12: Besides the missing significance test I suggest to change to % change.*

**Precipitation figures are now presented in percentage change and include stippling to illustrate regions that pass significance testing.**

*L313: "we have identified here with CESM1.2 in both of their models" make no sense.*

**Sentence rephrased as, "The same pronounced high-latitude warming and narrow region of tropical cooling identified in our study with CESM was also found by Kleinen et al. (2014) in both an EMIC (CLIMBER2) and an AOGCM (CCSM3), suggesting a signal robust to various climate models."**

*L330: You need to say that baroclinicity is changed in the lower part of the troposphere. In the upper part (say around 300 hPa) my guess is that the meridional temp. gradient is increased and thus baroclinicity is increased. So the vertical structure matters!.*

**Rephrased as, "Both inter-hemispheric and intra-hemispheric changes to the insolation gradient have roles to play, with the relatively high obliquity conditions of the early-mid MIS-11 interglacial (ca. 423-408 ka) a chief contributor to the lower-tropospheric low-baroclinicity conditions in boreal summer." Clarification will also be added that we refer to "baroclinicity" exclusively in the lower troposphere in our study.**

*L358:_ Here you can mention the Merz et al. (2014) publication you assess the effect of lapse rate, albedo changes etc. for the last Interglacial.*

**We will add the appropriate reference here, which fits nicely our discussion in the following sentence regarding surface temperature increases.**

*Review #2 (Alexander Robinson)*

*A key weakness of the study is that the Greenland ice sheet is prescribed to its present day configuration. However, this seems necessary and reasonable for the authors to be able to explore the phenomena of interest. Furthermore, this weakness is addressed well in the text. Nonetheless, I think there is an excellent opportunity to compare the quasi-time-series presented in Fig. 1 (JJA Greenland temps) with those of Robinson et al. (2017). There, peak summer warming around Greenland was estimated to be around 2-3degC, which is quite consistent with that presented here - obtained only by running a climate model with the right orbital configuration and GHGs. Making this comparison would add some value to the work here, and provide context for the promised future work with an interactive ice sheet component.*

**We would like to thank you for your constructive and supportive comments. We agree that incorporating the summer Greenland temperature time series developed by Robinson et al. (2017) into Figure 1 and our discussions will indeed bolster the work, providing some additional context and support for our results.**

*Some further minor points are listed below:*

*L79: Seems strange to have this posed as a question here. Perhaps rephrase into a sentence, or prepare the reader that you will ask a question.*

**Rephrased as, "In particular, our interest lies in identifying the atmospheric changes across the North Atlantic sector that were most consequential for mass balance changes in the Greenland ice sheet."**

*L285: of Hadley cell => of the Hadley cell*

**Accepted.**

*L286: for the tropical convection upon => for tropical convection on*

**Accepted.**